# EmbedMol: An Open Billion-scale Molecular Embedding Dataset for Molecular Discovery

## Abstract

Modern molecular libraries span billions of compounds, exposing a mismatch between dataset scale and the practicality of vHTS. SMILES strings remain the dominant representation, but while easy to store, they are difficult to consume at billion scale: each search or training run must first translate SMILES into learned features, incurring prohibitive overhead. We introduce *EmbedMol*, the first open billion-scale dataset of precomputed molecular embeddings, along with a scalable generation pipeline. *EmbedMol* comprises 977M embeddings from GDB13 and 11.2B embeddings from GDB13+ZINC22, generated with a deep model pretrained on experimental binding assays. Our contribution is not a new encoder, but a benchmark/dataset resource that makes billion-scale embedding-based retrieval practical. We demonstrate that precomputed vectors act as a faithful, efficient proxy for expensive inference, yielding up to $37.3\times$ speedups versus classical fingerprints and $1.5\times$ versus re-running the encoder, while maintaining strong retrieval quality across multiple targets. Beyond efficiency, *EmbedMol* establishes a testbed for billion-scale evaluation of retrieval methods, scaling behavior, and cross-target generalization in molecular discovery. To support reproducibility and accessibility, we release not only the dataset and loaders but also a fully automated AWS-based pipeline, enabling researchers with varying levels of distributed-systems expertise to reproduce and extend *EmbedMol*.

## 1 Introduction

Molecular datasets play crucial roles in modern molecular discovery tasks. Scanning through large chemical spaces using virtual high-throughput screening (vHTS) techniques allows researchers to identify molecules with desirable properties while navigating restrictions such as patents and application-specific requirements (Zhou and Zhong, 2017; Seifert et al., 2003; Cereto-Massagué et al., 2015; Hoffmann and Gastreich, 2019). For effective molecular screening, these datasets must be both voluminous and diverse, increasing the odds of uncovering molecules with optimal therapeutic or biological effects, while also being curated in formats that enable efficient processing (Hoffmann and Gastreich, 2019).

Yet as molecular libraries expand to billions or even trillions of compounds, a *mismatch* has emerged between dataset scale and the practicality of existing tools for vHTS (Stokes et al., 2020; Wong et al., 2024; Gilmer et al., 2017; Gasteiger et al., 2022b;a; Fang et al., 2022; Wang et al., 2020). On the one hand, these advanced tools heavily rely on large deep learning (DL) models to achieve high prediction accuracy, but this comes at the cost of increased computing time. For example, SOTA DL-based molecular discovery tools can only handle hundreds of millions of molecules ($10^8$) within a reasonable timeframe — typically days (Stokes et al., 2020; Neumann, 2022). On the other hand, current datasets are reaching sizes in the trillions ($\approx 10^{12}$). For example, GDB17 (Ruddigkeit et al., 2012) features 166 billion molecules ($1.66 \times 10^{11}$), and the most recent release of commercial datasets for off-shelf chemical spaces, such as eXplore from eMolecules, curates over 7 trillion ($\approx 10^{12}$) molecules purchasable on request (Neumann, 2022), as illustrated in Figure 1. Meanwhile, modern open datasets continue to grow in size, further exacerbating the mismatch and thus diminishing the usability of modern datasets. As a result, researchers and practitioners frequently resort to manually curating and trimming datasets to manage their scope effectively. For instance, leading solutions have reduced the ZINC15 dataset, initially comprising over a billion molecules ($10^9$), to 107 million molecules, by as much as 90% (Stokes et al., 2020; Sterling and Irwin, 2015). This approach requires

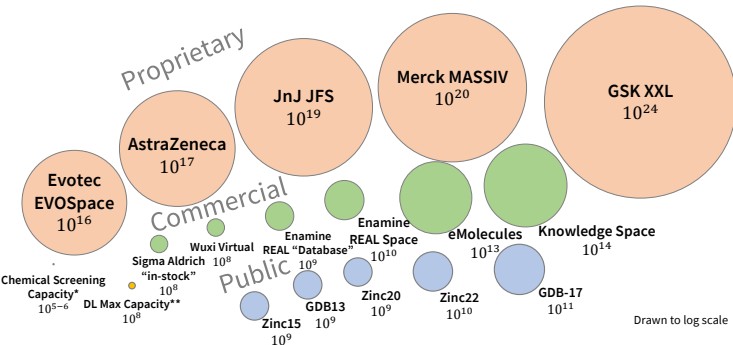

Figure 1: Existing ultralarge chemical datasets (Hoffmann and Gastreich, 2019; Sterling and Irwin, 2015; Tingle et al., 2023; Ruddigkeit et al., 2012; Blum and Reymond, 2009; Neumann, 2022; rea, 2024a;b; vir, 2024; mcu, 2024)

significant manual efforts to pinpoint specific chemical tranches that might harbor molecules with desirable physicochemical properties. More critically, it fails to leverage the extensive diversity and volume advantages of modern molecular datasets, thus hindering the discovery of ideal molecules in tranches beyond our present understanding.

One key reason behind such a mismatch is the *format* of how these molecules are stored within these chemical/molecular spaces, which is inefficient for searching and processing. Predominantly, open ultra-scale datasets for vHTS encode molecules as SMILES strings, a concise textual notation (Blum and Reymond, 2009; Ruddigkeit et al., 2012; Tingle et al., 2023; Sterling and Irwin, 2015). The textual representation of SMILES alone, however, does not directly encapsulate substructural or physicochemical details to be compared directly (Kim et al., 2021), so models for vHTS must first learn those molecular features and characteristics at high computational cost. Alternatively, researchers increasingly rely on latent-space molecular embeddings ("data-driven fingerprints"), the chemical analogue of feature vectors that drive large-scale image search (Fifty et al., 2023; Kim et al., 2021). For example, many existing works today use variational autoencoders (VAE) and other advanced DL techniques to perform computer-aided molecular design because their statistically learned representations preserve a rich set of critical features of molecules (Gao et al., 2020; Prykhodko et al., 2019; Bai et al., 2021; Gómez-Bombarelli et al., 2018; Lim et al., 2018; Ji et al., 2022; Pinheiro et al., 2022). This limitation is not only a practical barrier for chemists but also a methodological bottleneck for machine learning research. **The difficulty of consuming SMILES at scale prevents systematic study of representation quality, scaling behavior, and cross-target generalization, which are key questions for the ML community.**

Unfortunately, despite the continuous growth of molecular datasets and the increasing interest in data-driven fingerprints, *none of these datasets feature molecules in statistically learned embeddings*. Two primary factors contribute to this gap. For molecular researchers focused on vHTS outcomes, embeddings are often treated as incidental byproducts of DL models, leading to a general lack of deliberate curation and making them difficult to access and utilize. In contrast, computing experts tend to prioritize the design of embedding generation algorithms over the systematic curation of these embeddings. In both cases, the challenge is compounded by the high technical requirements needed to generate them, requiring advanced expertise in neural network architectures, molecular sciences, and distributed computing frameworks, all at the same time.

To address the challenges associated with early-stage vHTS in computational molecular discovery and to facilitate greater accessibility of latent-space molecular representations for researchers, we assembled a publicly available embedding dataset comprising high-quality molecular embeddings derived from a SOTA DL-based molecular property prediction model. Specifically, we introduce *EmbedMol*, the first and largest publicly available dataset of data-driven molecular embeddings tailored explicitly for vHTS, encompassing over 11 billion molecules aggregated from the ZINC22 and GDB13 databases Tingle et al. (2023); Blum and Reymond (2009). By focusing on enhancing the critical initial stages of the molecular discovery pipeline, *EmbedMol* significantly improves the efficiency of existing vHTS workflows, enabling more precise navigation of vast chemical spaces for molecules with desirable properties. Although primarily intended for vHTS applications, the versatility of *EmbedMol* also supports potential expansions into various downstream tasks, which we leave open for future exploration. **Our contribution is not a new encoder but a benchmark/dataset**

**resource that makes billion-scale retrieval practical and reproducible.** We validate EmbedMol along three axes: (i) embedding quality for high-precision vHTS, (ii) cross-target generalizability, and (iii) deployment efficiency at billion scale. By releasing both the dataset and the pipeline, we lower the barrier for researchers with varying levels of distributed-systems expertise and establish a foundation for future ML directions such as multimodal integration, transfer learning, and large-scale benchmarking across embedding families.

To enable reproducibility and lower technical barriers, we have built—and are open-sourcing—a custom software tool that leverages the advanced public cloud computing services provided by Amazon Web Services (AWS). This pipeline automates dataset generation using the platform's parallelism and flexibility, allowing researchers without distributed-systems expertise to reproduce and extend *EmbedMol*.

In our evaluation, *EmbedMol* achieves a retrieval precision of up to 68% among the top-100 candidates on a 3.5B-molecule subsample, using DL-predicted drug-target binding affinity as ground truth—performance comparable to SOTA embedding datasets in recent large-scale retrieval benchmarks (Wong et al., 2024; Simhadri et al., 2022). Cross-target experiments show consistent, high-quality retrieval without retraining, and approximate nearest-neighbor search over *EmbedMol* improves vHTS throughput by up to $37.3\times$ versus non-DL fingerprints and $1.5\times$ relative to re-running the source DL encoder. Together, these results confirm that precomputed embeddings act as a faithful and efficient proxy for expensive inference, enabling billion-scale screening that was previously impractical.

In summary, our work makes three contributions: **(i)** we release *EmbedMol*, the first open billion-scale dataset of precomputed molecular embeddings (11B molecules from GDB13 and ZINC22) with a scalable cloud pipeline; **(ii)** we validate its utility across embedding quality, cross-target generalizability, and deployment efficiency; and **(iii)** we provide tools that lower barriers for researchers with varying systems expertise, enabling future work on multimodal integration, transfer learning, and large-scale benchmarking.

The remainder of the paper reviews related datasets (§ 2), presents our collection pipeline (§ 3), describes dataset composition (§ 4), and reports evaluation results (§§ 5 and 6), before concluding with discussion and access details (§§ 7 and 9).

## 2 BACKGROUND

The molecular datasets for computational molecular discovery have experienced a significant upswing over the past decade in size and diversity but little in curation format. In this section, we review the existing molecular datasets based on these three requirements: size, diversity, and format.

**Size** Making large molecular datasets has become a collective effort from academia and industry (Hoffmann and Gastreich, 2019; Sterling and Irwin, 2015; Tingle et al., 2023; rea, 2024a;b; vir, 2024; mcu, 2024), as shown in Figure 1. The most prominent examples from academia for open molecular datasets are the Generic Databases (GDB). Its newest version, GDB17, features 166 billion molecules with up to 17 atoms (Ruddigkeit et al., 2012). It is one of the largest open datasets generated through molecular enumeration. By enumerating all possible atom combinations of organic molecules under physical and chemical constraints, GDB17 is set to explore and better define known and unknown chemical space, offering molecular researchers and practitioners millions of isomers of known drugs and analogs that retain high shape similarity. Other academic efforts also focus on developing large datasets for tangible compounds from existing molecule suppliers' purchasable drug catalogs. ZINC22, the newest version of such, contains over 37 billion commercially available compounds sourced from Enamine (REAL) (rea, 2024a;b), WuXi (GalaXi) (vir, 2024), and Mcule (Ultimate) (mcu, 2024; Tingle et al., 2023).

Major chemical suppliers such as Enamine, Wuxi, and OTAVA have collectively generated datasets covering over 50 billion synthesizable and purchasable molecules. SOTA commercial molecular spaces, such as the eMolecular and eXplore datasets, now encompass an estimated 7.0 trillion compounds generated using 49 robust chemical reactions, representing the most comprehensive commercially available molecular space to date (Neumann, 2022). In the industry, companies are developing even larger proprietary chemical spaces: Merck's MASSIV dataset contains up to $10^{20}$

molecules, while GSK's GSK XXL dataset includes $10^{26}$ molecules, placing them at the leading edge of the field (Hoffmann and Gastreich, 2019).

**Diversity** Researchers commonly use two techniques—enumeration and combinatorics—to explore chemical space and diversify molecular datasets. Enumeration involves exhaustively generating all unsaturated hydrocarbon skeletons and substituting atoms to create potential molecular candidates (Blum and Reymond, 2009). The GDB datasets, for example, are built using this approach. In this process, all possible mathematical graphs meeting basic molecular criteria are selected and expanded into hydrocarbon skeletons, with nodes as atoms and edges as chemical bonds. These skeletons are then converted into molecules by replacing edges with covalent bonds and nodes with heavy atoms, followed by filtering candidates based on valency rules, chemical stability, and functional group constraints.

Combinatorial chemistry offers another route for dataset generation (Boehm et al., 2008). Researchers use chemical fragments (such as amino acids or spiro-compounds) as building blocks, assembling new compounds by applying known reaction rules. This allows chemists to diversify the chemical classes and ensure the synthesizability of the generated molecules.

**Format** Traditionally, molecules are represented in these datasets by a textual format known as SMILE, or Simplified Molecular Input Line Entry System, which does not include crucial substructural information. However, converting these SMILE strings to 2D or 3D molecular structures useful for *in silico* discovery tasks involves some non-trivial computation. Hence, research has focused on developing numerical vectors that directly encapsulate substructural details and other molecular characteristics to enhance input quality and better represent the molecule's properties. Rule-based fingerprints such as Morgan (Morgan, 1965), ECFP (Rogers and Hahn, 2010), MACCS (Durant et al., 2002), and Daylight (Daylight, 2024) represent fingerprints that are constructed based on set rules created by domain experts based on learned or empirical knowledge of molecules extracted from previous research. These fingerprints are usually presented as binary bit-strings, with each bit representing key substructures or key connectivity path fragments.

The research community is increasingly adopting statistically derived latent-space molecular representations—also known as data-driven fingerprints—to enhance the informativeness of molecular data. These fingerprints are generated using deep learning architectures such as autoencoders (AEs), recurrent neural networks (RNNs), and graph neural networks (GNNs), which learn continuous latent representations of molecules. Numerous studies have shown that these embeddings outperform traditional rule-based fingerprints, owing to the strengths of statistical learning (Gao et al., 2020; Prykhodko et al., 2019; Bai et al., 2021; Gómez-Bombarelli et al., 2018; Lim et al., 2018; Fifty et al., 2023; Kim et al., 2021). Despite their strong representational capabilities, data-driven fingerprints (embeddings) rarely curated or stored for reuse, requiring researchers to repeatedly regenerate them—a redundant and time-consuming process. For molecular researchers focused on vHTS outcomes, extra time and effort must be spent on generating embeddings rather than directly analyzing molecular activity. Meanwhile, computing experts tend to concentrate on developing new embedding generation algorithms rather than on establishing robust curation practices. Together, these factors hinder the broader adoption and efficient use of statistically learned embeddings in molecular research.

While numerous learned encoders have been proposed (e.g., graph-based and sequence-based models), none are available at billion scale as ready-to-use vectors. This absence prevents fair, apples-to-apples comparison of retrieval methods and scaling behavior across representation families, motivating the need for an open, large-scale embedding resource such as *EmbedMol*.

## 3 COLLECTION METHODOLOGY

Given our interest in statistically derived embedding representations of molecules, we adapted existing DL models to extract latent molecular representations within the DL network pipeline. At a high level, the embedding collection pipeline has three main components: (i) downloading and preprocessing of SMILE strings, (ii) distributed embedding generation, and (iii) molecule filtering. We implement this pipeline on the cloud platform using tools from the Amazon EC2 instances. The details of each component are described as follows.

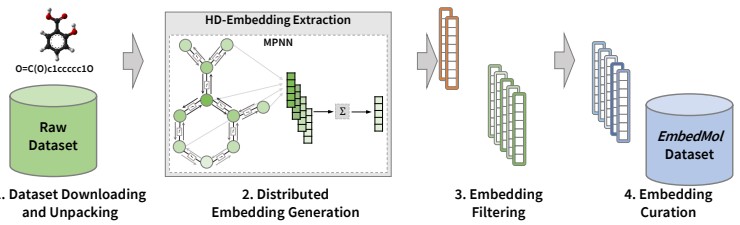

Figure 2: *EmbedMol* similarity search pipeline

**SMILE String Downloading** In line with our focus on ultra-large-scale molecular datasets, we generate our embedding using two popular billion-scale molecular datasets that use SMILE encoding. These datasets were specifically chosen not only for their large size but also for their wide accessibility in the academic and scientific communities. Both datasets categorize molecules into groups based on distinct criteria, with individual SMILES representations for each molecule.

- **GDB13** comprises a comprehensive library of small organic molecules, each containing no more than 13 atoms of C, N, O, S, and Cl. Generated through a process based on fundamental chemical stability and synthetic feasibility rules, it encompasses approximately 970 million molecules suitable for virtual drug screening (Blum and Reymond, 2009).

- **ZINC22** datasets contain an ultra-size collection of about 37 billion make-on-demand compounds in SMILE representations (Tingle et al., 2023). As an open dataset, ZINC22 is freely available via the GUI website cartblanche22.docking.org.

The selected datasets are first downloaded, unzipped, and processed asynchronously using Python thread libraries. These works are performed using throughput and I/O provisioned EBS volume as the backing storage, allowing for maximized utilization of parallelism in the cloud node's processing power and network bandwidth. After data cleansing, these datasets are stored in the original text format, with additional annotations removed. To enable data-level parallelism in an embedding generation, these molecular representations are then batched into chunks, with roughly 25 million in each batch, for processing in the next stage of the embedding generation pipeline.

**Embedding Generation** We generate embeddings by exposing the compound encoder of a strong, publicly available DTI model (Huang et al., 2020), so the network emits both the latent compound vector and the final prediction. This preserves the backbone's predictive fidelity while yielding a reusable representation; critically, the backbone was pretrained on experimentally measured binding assays, grounding the embeddings in empirical data. Our modification is architecture-agnostic for encoder-decoder models and does not change prediction quality; it simply surfaces the encoder output at scale.

We choose DeepPurpose as our backing DL library for four reasons: We choose DeepPurpose as our backing DL library for four reasons. **Architecture:** its modular design clearly separates compound encoders from prediction decoders, simplifying embedding extraction. **Open source:** the library is fully open, enhancing reproducibility and accessibility. **Broad pre-training:** it is pretrained on diverse molecular tasks (DTI, property prediction, PPI, and protein function), producing robust embeddings for vHTS use cases such as repurposing and QSAR. **Input format:** unlike many SOTA models that require 3D structural data (Gasteiger et al., 2022b;a; Fifty et al., 2023; Fang et al., 2022), DeepPurpose operates directly on SMILES strings, making it especially practical for billion-scale datasets where only SMILES strings is available.

For this version of *EmbedMol*, we select a message-passing neural network (MPNN) encoder, as MPNNs explicitly model inter-atom interactions and capture key spatial information. Their effectiveness for molecular representation learning has been repeatedly demonstrated in benchmarks against multiple SOTA baselines (Gasteiger et al., 2022b;a; Fang et al., 2022).

**Compute Automation** To maximize resource utilization and enable *data-level parallelism* during embedding generation, we use a small dataset to perform micro-benchmarks on existing Amazon EC2 instances. Guided by our micro-benchmark result, we chose a medium-sized Intel Sapphire Rapid cloud node with 48 vCPUs and 128 GiB of RAM, with extra provisioned IOPS and bandwidth storage mediums, as a worker instance through Amazon Elastic Block Store. These worker nodes can process 25 million molecules in 1 day when three DL models run parallel on the same worker instance. Using AWS's AMI and EC2 launch template, we were able to spawn roughly 40 workers in

Table 1: Dataset statistics

| Source Dataset | Dimension | Count |
|----------------|-----------|-------|
| *EmbedMol*-1B | $1 \times 128$ | 977,468,314 |
| *EmbedMol*-11B | $1 \times 128$ | 11,231,133,850 |

parallel to automatically process all 11 billion molecules in multiple rounds using the aforementioned chunks of molecules without per node-specific configuration. We also utilize EC2 snapshots every 24 hours to facilitate recovery from potential generation pipeline failure.

**SMILE String Filtering** During the embedding generation of the 11 billion molecules, we identified several invalid SMILE strings that could not be interpreted into actual molecules using the RDKit (v2023.09.3) library. Most of these molecules fail to be interpreted into actual molecules due to the illegal canonical SMILE string format. We flagged these molecules during our embedding generation process. Subsequently, we removed them from our embedding dataset during evaluation as they did not result in valid high-dimensional embeddings. This process removes less than 0.01% of the original datasets and hence does not impact the variety and scale of our datasets. We leave more fine-grained and detailed validation of these molecules to future work out of the scope of this paper.

## 4 DATASET COMPOSITION

We release two datasets: *EmbedMol*-1B, comprising 977M embeddings derived solely from GDB13, and *EmbedMol*-11B, comprising 11B embeddings from GDB13 and ZINC22 for molecules up to 26 heavy atoms (Table 1). Each dataset is partitioned into batches and stored as compressed NumPy arrays, with metadata including the SMILES string, dataset source, and an RDKit validity flag (1/0). Invalid molecules, as described in § 3, can be removed by filtering on this flag.

## 5 EVALUATION OBJECTIVE AND SETUP

We construct an end-to-end molecular discovery pipeline using similarity search, similar to the ones used in (Zhu et al., 2020a) to validate the performance and quality of our generated embeddings. Note that the similarity search-based method is widely used in current molecular discovery research (Cereto-Massagué et al., 2015), enabling us to assess our dataset in real-world application scenarios. In particular, we employ the FAISS framework for similarity searches, an open-source C++ and Python library adept at similarity search and clustering for dense vectors (Douze et al., 2024; Johnson et al., 2017). This framework is ideal for handling extensive vector sets within physical memory limits, as with our multi-billion molecule dataset. Although GPUs are increasingly used in ML, we focus on CPU-based similarity search, as CPUs remain the most accessible resource for domain scientists and require no specialized expertise. Our methods extend to GPUs, but we leave such exploration for future work. All experiments are conducted on AWS EC2 instances equipped with memory-optimized Sapphire Rapids CPUs. Most of our evaluations utilize the *EmbedMol*-1B dataset, with additional experiments performed on a 3.5 billion compound random subsample of the *EmbedMol*-11B dataset.

**Evaluation scope** We do not claim to outperform all learned embeddings. Our goals are (i) to test whether precomputed *EmbedMol* vectors act as a faithful proxy for their source model at billion scale, and (ii) to compare against de facto baselines (fingerprints) that remain standard in ultra-large libraries. Exhaustive cross-architecture comparisons and wet-lab assays are out of scope here and are enabled by releasing this resource.

**Algorithm Selection** While indexing algorithms range from hashing-based to tree-based and graph-based indices, for *EmbedMol*'s initial version, we focus on the Inverted File Product Quantization (IVFPQ) index. IVFPQ caters to platforms with limited memory, which is especially suitable for our ultra-large billion-scale molecular datasets. All indices are trained on a randomly subsampled training set of all molecular embeddings from our embedding dataset, which comprises 0.02% of the original dataset. We provide details regarding the hyperparameters used § A. (see § B for sensitivity analysis)

Table 2: Fingerprints used for the baseline experiments and their generation parameter

| Fingerprints Name | Generation Parameters | Bit Width |
|---|---|---|
| RDKit Daylight | $minPath = 1, maxPath = 7, bitsPerHash = 2, hash = true$ | 2048-bit |
| MACCS | RDKit implementation of MACSS 166-bit version | 166-bit |
| Morgan3 | $radius = 3$ | 1024-bit |

# 6    EXPERIMENTAL RESULTS

Our evaluation proceeds along three axes: (i) *embedding quality*, testing whether *EmbedMol* provides sufficient representational power for high-precision vHTS relative to baselines; (ii) *cross-target generalizability*, examining whether the embeddings capture global molecular properties beyond a single search target; and (iii) *deployment efficiency*, quantifying the computational and time savings that make ultra-large-scale vHTS feasible in practice.

## 6.1    DATASET QUALITY FOR VHTS

Rather than measuring *EmbedMol*'s ability to retrieve structurally similar molecules, we focus on a more practical, real-world scenario by evaluating the precision in retrieving molecules predicted to possess desirable properties of our dataset. Concretely, the evaluation process largely resembles a vHTS task in molecular discovery: searching for the top antiviral candidates for the COVID-19 target 3CL protease. The SARS-CoV-2 3CL protease (also known as the main protease) plays a crucial role in the viral life cycle and is a key target for small-molecule COVID-19 therapy (Liu et al., 2022; Zhu et al., 2020b; Boras et al., 2021). The ground truth is the predicted binding affinity score between the *EmbedMol*-1B dataset's molecules and the COVID-19 protease. As outlined in § 1 and § 2, existing ultra-scale molecular datasets are typically generated through enumeration or combinatorial techniques, resulting in a general lack of empirical property data. A common approach is to use DL models to predict empirical properties for curated molecules. These predictions serve as our best estimates of key molecular properties and are treated as ground truth in our experiments.

To ensure a fair evaluation, we use separate encoders for drug and protein targets when establishing ground truth, distinct from the encoder used to generate embeddings. This prevents performance bias from relying on the same DL backbone. Additionally, we report retrieval accuracy at various top candidate levels, following established standards in the information retrieval community (Zhu et al., 2022; noa, 2024). Focusing on retrieval accuracy provides a clear and widely accepted measure of performance.

For traditional vHTS using handcrafted fingerprints, we have chosen 3 widely used and representative fingerprint types and 2 similarity metrics. For the 2 similarity metrics, we choose the most widely used Tamimoto similarity (Jaccard, 1901; Tanimoto, 1958) and Braun-Blanquet similarity (noa, 1994; Braun-Blanquet et al., 1932) following benchmarks made in (Safizadeh et al., 2021). For the 3 fingerprints (Daylight, 2024; Durant et al., 2002; Rogers and Hahn, 2010; Morgan, 1965), detailed generation parameters are provided in Table 2 [1]. These options rank among the best-performing fingerprints in vHTS today, offering a balanced blend of structural and hashed fingerprint approaches. Among these, the Morgan fingerprints are also identified in the prior literature as the best-performing fingerprints across diverse applications, including small-molecule virtual screening (O'Boyle and Sayle, 2016; Riniker and Landrum, 2013; Glem et al., 2006). For these fingerprints and similarity metrics, we assume the best possible results by using brute-force methods to identify the exact top candidates; these results are marked as BF in Table 3.

We capture the performance of the *EmbedMol* in COVID-19 antiviral drug hit identification, along with the baseline performances of several existing and widely used fingerprint-based vHTS techniques on the same workload in Table 3. Our approach achieves a retrieval precision of 31% among the top 100 candidates in the smaller *EmbedMol*-1B dataset. Applying the same experiment to a 3.5 billion subset of *EmbedMol*-11B,which offers greater molecular diversity,yields an even higher precision:

---

[1]For Morgan/ECFP baselines, we follow established best practice and use 1024-bit radius-3 fingerprints (Morgan3). Prior work has shown that longer bit vectors yield diminishing returns, with 1024-bit vectors strongly correlating with 16K-bit versions ($r^2 > 0.99$) (Riniker and Landrum, 2013). Our own microbenchmarks confirm this principle: increasing to 2048 bits produced no measurable precision gain while incurring higher memory and compute costs.

Table 3: The performance of *EmbedMol* in drug hit identification

| Model | Embedding Type | Methods | Prec@5 | Prec@50 | Prec@100 |
|---|---|---|---|---|---|
| **COVID-19 Protease** (Liu et al., 2022; Zhu et al., 2020b; Boras et al., 2021) | | | | | |
| *EmbedMol*-1B | RDKit-Daylight | Tamimoto (BF) | 20% | 4% | 2% |
| | | Braun-Blanquet (BF) | 20% | 4% | 2% |
| | MACCS | Tamimoto (BF) | 20% | 4% | 2% |
| | | Braun-Blanquet (BF) | 20% | 4% | 2% |
| | Morgan3 | Tamimoto (BF) | 20% | 2% | 1% |
| | | Braun-Blanquet (BF) | 20% | 2% | 2% |
| | *EmbedMol* | IVFPQ | 20% | 24% | 31% |
| *EmbedMol*-11B (3.5B Subsample) | *EmbedMol* | IVFPQ | 40% | 68% | 54% |
| **COVID-19 Helicase** (Halma et al., 2022; Otsuka et al., 2024; Knany et al., 2023) | | | | | |
| *EmbedMol*-1B | RDKit-Daylight | Tamimoto (BF) | 20% | 4% | 2% |
| | | Braun-Blanquet (BF) | 20% | 4% | 2% |
| | MACCS | Tamimoto (BF) | 20% | 4% | 2% |
| | | Braun-Blanquet (BF) | 20% | 4% | 2% |
| | Morgan3 | Tamimoto (BF) | 20% | 2% | 1% |
| | | Braun-Blanquet (BF) | 20% | 2% | 2% |
| | *EmbedMol* | IVFPQ | 20% | 24% | 31% |
| **MMP9** (Mondal et al., 2020; Yabluchanskiy et al., 2013; mmp, 2025) | | | | | |
| *EmbedMol*-1B | *EmbedMol* | IVFPQ | 40% | 26% | 32% |
| **LCK** (Chiang and Hodes, 2016; Laukkanen et al., 2022; Palacios and Weiss, 2004) | | | | | |
| *EmbedMol*-1B | *EmbedMol* | IVFPQ | 20% | 42% | 28% |

Table 4: Performance comparison of non-DL based vHTS

| Source Dataset | Method | Build Time | Avg. Query Time |
|---|---|---|---|
| *EmbedMol*-1B | IVFPQ | 1.83 hrs | $0.13 \pm 0.06$ sec |
| *EmbedMol*-1B | Brute-force | n/a | $2.41 \pm 1.58$ hrs |
| GDB13 (Baseline) | Brute-force | 88.5 hrs | $1.42 \pm 0.83$ hrs |

68% for the top 50 candidates and 54% for the top 100 candidates. These results, obtained with minimal index building and parameter tuning, contrast sharply with standard vHTS methods based on handcrafted fingerprints, which typically exhibit single-digit precision for the same retrieval ranges.

## 6.2 CROSS-TARGET GENERALIZABILITY OF *EmbedMol*

To assess the usability of the generated embeddings for drug target binding prediction in molecular discovery, we perform similarity searches on these molecules for vHTS against binding targets not included during embedding generation. Specifically, we evaluate precision by identifying top candidate molecules for three distinct targets: COVID-19 Helicase, MMP9 (Mondal et al., 2020; Yabluchanskiy et al., 2013; mmp, 2025), and LCK (Chiang and Hodes, 2016; Laukkanen et al., 2022; Palacios and Weiss, 2004)—all of which are high-profile and relevant targets for drug discovery, with Helicase serving as a known target for inhibiting viral replication (Halma et al., 2022; Otsuka et al., 2024; Knany et al., 2023). We reuse the best-performing high-dimensional vector indices constructed for *EmbedMol*-1B in earlier research questions, directly applying them to similarity searches for these new binding targets. As a result, we achieved a consistent retrieval precision of about 30% on the top 100 candidate molecules for all molecular binding targets. Comparative performance metrics between our method and established vHTS baselines are reported in table 3. For a fair comparison, we use identical fingerprint parameters for all baseline methods without additional tuning.

## 6.3 TIME-SAVINGS ADVANTAGE FOR VHTS

The central challenge addressed by *EmbedMol* is the high computational cost of converting molecules from SMILES (or other textual formats) into embeddings. By providing precomputed *EmbedMol* embeddings, we aim to accelerate both deep learning and traditional fingerprint-based vHTS during training and inference. In this experiment, we evaluate the potential time savings for DL-based and non-DL-based vHTS workflows when using our dataset at search time. By relying on these

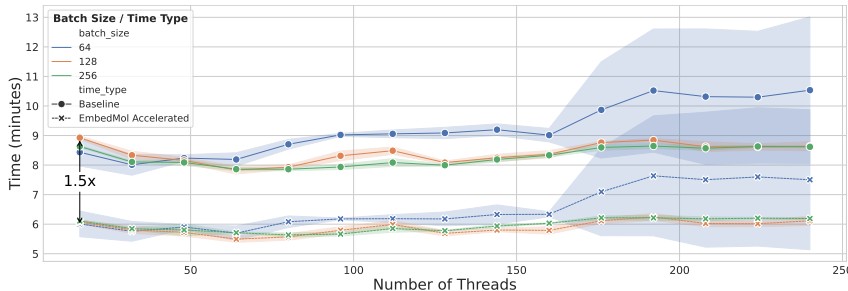

Figure 3: Execution time vs. threads for baseline and accelerated methods utilizing *EmbedMol*

embeddings, practitioners can bypass fingerprint generation and representation learning steps, greatly improving the speed and efficiency of vHTS on ultra-large datasets.

We compare two scenarios: (1) traditional similarity-search-based (non-DL) and (2) property-prediction-based (DL) vHTS, measuring total runtime on 15 search targets. In the baseline, only SMILES are available. With *EmbedMol*, users have access to both embeddings and SMILES. All experiments use the *EmbedMol*-1B dataset, with standard optimizations like caching intermediate results and pre-generating fingerprints for reuse employed.

For the traditional non-DL vHTS methods, we assume an optimal pipeline where the top 100 candidates are retrieved with a single database scan (Table 4). In all brute-force tests on 100,000 randomly selected molecules, *EmbedMol* achieves up to a 37.3× speed-up per search by eliminating the need for fingerprint generation. It matches baseline methods in average query time if cached fingerprints are reused—though this is not yet standard practice. Using approximate similarity search on both GDB13 and *EmbedMol*-1B (as in (Zhu et al., 2020a)) and a 1.83-hour index-building phase, per-search costs drop to under a second, while maintaining the high retrieval precision shown in earlier results.

For DL-based property prediction, we remove the SMILES translation backbone so embeddings can be used directly for inference. We measure total inference time for 100,000 randomly selected molecules (the same subset as above) using the original drug-target interaction model, modified only to accept embeddings as input. Model validity is confirmed by comparing predictions of the modified and original architectures. As shown in Figure 3, varying the number of threads, we observe a consistent 1.5× improvement in vHTS throughput across all levels of parallelism.

## 7 CONCLUSIONS

We presented *EmbedMol*, the first open billion-scale molecular embedding dataset and a scalable generation pipeline. Precomputed vectors provide an efficient, high-fidelity proxy for expensive inference, enabling billion-scale retrieval with up to 37× speedups over classical baselines while maintaining strong quality. The contribution is a benchmark/dataset resource—not a new encoder—that establishes a foundation for large-scale research on scaling laws, cross-target transfer, and multimodal integration in molecular representation learning. We release the dataset, loaders, and pipeline to catalyze community work that was previously computationally prohibitive. See Reproducibility Statement for dataset access and licensing.

## 8 LIMITATIONS

Our goal is to release a curated billion-scale embedding dataset and pipeline, not a new model, so we rely on strong existing architectures for embedding generation. As with all billion-scale vHTS, exhaustive experimental validation is infeasible; evaluation therefore relies on predicted ground truth, underscoring the utility of embeddings that can prioritize limited assays. While this bounds our claims, it motivates future benchmarks now enabled by *EmbedMol*, including (i) comparisons across learned embedding families (e.g., chemical language models), (ii) multimodal fusion with protein sequence/structure text, and (iii) targeted experimental validation for high-confidence hits.

## 9 REPRODUCIBILITY STATEMENT

For review purposes, *EmbedMol* is temporarily hosted on Dryad at `http://datadryad.org/share/LINK_NOT_FOR_PUBLICATION/` `KCFD7gIfpEBCabATHI1mg7aOcY4jWUc9-JRIdJ02fYk`, with a mirror at `http://datadryad.org/share/LINK_NOT_FOR_PUBLICATION/jPViJsUAM_` `8AI7uSnQfCoQ-7APLiVswEf8P3xk-eo_0`. Upon acceptance, the dataset will be permanently archived and maintained on a long-term public repository (link to be provided after review). Usage of the source datasets, as well as distribution of the resulting high-dimensional vectors, follows the terms set by the original dataset providers. Restrictions on redistribution and patent use are mandated by those agreements, not by us as authors (see original dataset licenses in Appendix C). While *EmbedMol* is freely available for download, it must not be used in patents, nor redistributed—either wholly or in substantial portions—without explicit written consent from the corresponding author.

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

## A    APPROXIMATE SIMILARITY SEARCH INDEX PARAMETERS

In Table 5, we present the key parameters and statistics for all IVFPQ indices trained on the *EmbedMol*-1B and 11B (3.5B subsample) datasets. For each dataset, we varied several IVFPQ configuration parameters, including the number of bits per sub-quantizer (**nbits**), the number of sub-quantizers (**m**), and the number of Voronoi cells (**nlist**), to examine their impact on index size, training time, addition time, and, most importantly, search performance. In this context, a Voronoi cell is a partition of the vector space, where each cell groups vectors closer to a given cluster center than any other. This clustering enables efficient grouping of similar vectors and restricts search operations to only the most relevant cells, greatly improving search speed on billion-scale datasets. This comprehensive benchmarking guides the selection of index parameters and their associated resource requirements. The table provides transparency and reproducibility for our indexing process, ensuring other researchers can replicate or adapt our approach for large-scale molecular similarity search tasks.

Table 5: Key statistics of trained IVFPQ indices across datasets

| Source Dataset | Index | nbits | m | nlist | Size | Training Time | Adding Time |
|---|---|---|---|---|---|---|---|
| *EmbedMol*-1B | IVFPQ | 8 | 2 | 20,000 | 8.9 GB | 33.2 s | 1.42 hrs |
| | IVFPQ | 8 | 4 | 20,000 | 11 GB | 33.3 s | 1.56 hrs |
| | IVFPQ | 8 | 8 | 20,000 | 15 GB | 33.6 s | 1.69 hrs |
| | IVFPQ | 8 | 16 | 20,000 | 22 GB | 33.9 s | 1.63 hrs |
| | IVFPQ | 8 | 32 | 20,000 | 36 GB | 34.8 s | 1.69 hrs |
| | IVFPQ | 8 | 64 | 20,000 | 64 GB | 36.5 s | 1.83 hrs |
| | IVFPQ | 2 | 8 | 20,000 | 8.9 GB | 32.8 s | 1.34 hrs |
| | IVFPQ | 4 | 8 | 20,000 | 11 GB | 32.5 s | 1.41 hrs |
| | IVFPQ | 6 | 8 | 20,000 | 13 GB | 32.9 s | 1.42 hrs |
| | IVFPQ | 8 | 8 | 8,000 | 15 GB | 14.0 s | 1.25 hrs |
| | IVFPQ | 8 | 8 | 9,000 | 15 GB | 15.6 s | 1.11 hrs |
| | IVFPQ | 8 | 8 | 10,000 | 15 GB | 17.2 s | 1.15 hrs |
| | IVFPQ | 8 | 8 | 30,000 | 15 GB | 49.9 s | 1.89 hrs |
| | IVFPQ | 8 | 8 | 40,000 | 15 GB | 61.9 s | 1.72 hrs |
| *EmbedMol*-11B (3.5B Subsample) | IVFPQ | 8 | 2 | 20,000 | 33 GB | 61.0 s | 8.15 hrs |
| | IVFPQ | 8 | 4 | 20,000 | 40 GB | 64.2 s | 8.48 hrs |
| | IVFPQ | 8 | 8 | 20,000 | 53 GB | 74.7 s | 9.58 hrs |
| | IVFPQ | 8 | 16 | 20,000 | 79 GB | 62.8 s | 8.50 hrs |
| | IVFPQ | 8 | 32 | 20,000 | 131 GB | 63.6 s | 9.59 hrs |
| | IVFPQ | 8 | 64 | 20,000 | 235 GB | 67.8 s | 8.75 hrs |
| | IVFPQ | 2 | 8 | 20,000 | 33 GB | 64.7 s | 7.95 hrs |
| | IVFPQ | 4 | 8 | 20,000 | 40 GB | 68.4 s | 8.71 hrs |
| | IVFPQ | 6 | 8 | 20,000 | 46 GB | 64.8 s | 7.78 hrs |
| | IVFPQ | 8 | 8 | 8,000 | 53 GB | 12.8 s | 7.72 hrs |
| | IVFPQ | 8 | 8 | 9,000 | 53 GB | 15.6 s | 7.95 hrs |
| | IVFPQ | 8 | 8 | 10,000 | 53 GB | 19.6 s | 8.15 hrs |
| | IVFPQ | 8 | 8 | 30,000 | 53 GB | 132.9 s | 8.49 hrs |
| | IVFPQ | 8 | 8 | 40,000 | 53 GB | 249.8 s | 7.88 hrs |

## B    SENSITIVITY TEST FOR *EmbedMol* ON TRAINED INDICES

We also perform a sensitivity test by evaluating the impact of index construction parameters on the precision of similarity search performance in molecular discovery tasks by investigating the behavior of the IVFPQ index under various parameter adjustments for both *EmbedMol*-1B and a 3.5-billion subsampled *EmbedMol*-11B. This is not featured in the main text of this paper. As shown in Figures 4 and 5, increasing the number of sub-quantizers from 2 to 64 in the IVFPQ index triples top-100 precision, while adjusting quantization and Voronoi cells yields improvements consistent with ANN theory.

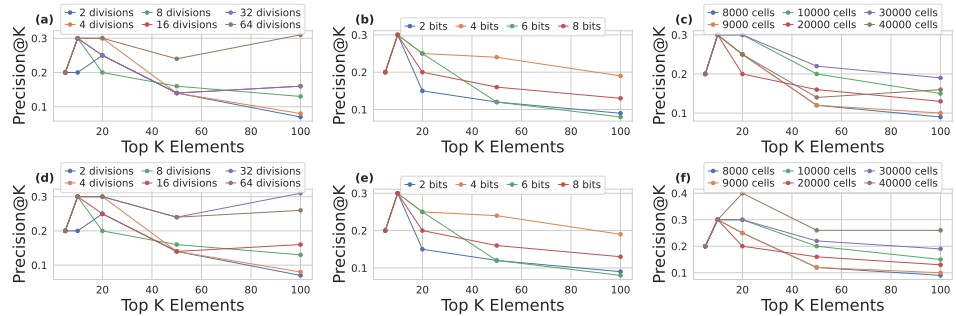

Figure 4: Precision of IVFPQ index on *EmbedMol*-1B in retrieving top antiviral candidate for SARS-CoV 3CL Protease (a-c) and SARS-CoV2 Helicase (d-f)

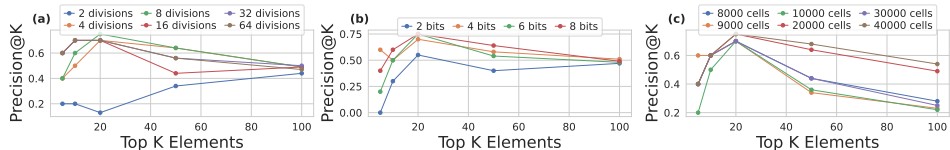

Figure 5: Precision of IVFPQ index on subsample of *EmbedMol*-11B retrieving top antiviral candidate for SARS-CoV 3CL Protease.

## C   ORIGINAL LICENSES FOR SOURCE DATASETS

We include below the license terms of GDB13 and ZINC22. These upstream conditions govern redistribution and patent restrictions in *EmbedMol*, as noted in our Reproducibility Statement.

### C.1   GDB13

> Terms and conditions: The GDB databases may be downloaded free of charge. In published research involving GDB, cite the appropriate references mentioned above. GDB must not be used as part of or in patents. GDB and large portions thereof must not be redistributed without the express written permission of Jean-Louis Reymond.

### C.2   ZINC22

> ZINC - 22 is free to use for everyone, but you may not redistribute major portions without the express written permission of John Irwin, chemistry4biology@gmail.com.

