# OpenReview forum: "EmbedMol: An Open Billion-scale Molecular Embedding Dataset for Molecular Discovery"
_ICLR.cc/2026/Conference — ICLR 2026 Conference Withdrawn Submission_

### Official Review · Reviewer_oZ81 · 2025-10-26

**Soundness:** 2
**Presentation:** 1
**Contribution:** 1
**Rating:** 2
**Confidence:** 4

**Summary:**

This paper introduces EmbedMol, a large-scale dataset of precomputed molecular embeddings derived from 11 billion molecules sourced from GDB13 and ZINC22. The authors position this as a resource to address the computational bottleneck of generating data-driven fingerprints for virtual high-throughput screening (vHTS). The work includes a scalable cloud pipeline for generating the embeddings and evaluates the dataset on retrieval precision, cross-target generalizability, and deployment efficiency.

**Strengths:**

Valuable Resource: The creation and public release of a billion-scale molecular embedding dataset is a significant undertaking with clear potential utility for the community. It lowers the barrier to entry for researchers without extensive distributed computing expertise to work with large-scale molecular representations.

Practical Focus: The paper addresses a genuine and pressing bottleneck in computational drug discovery: the infeasibility of running DL models on trillion-molecule libraries. Providing precomputed embeddings is a pragmatic solution.

Reproducibility and Engineering Effort: The release of the dataset, along with a detailed description of the scalable cloud pipeline, promotes reproducibility and demonstrates a substantial engineering effort.

**Weaknesses:**

1. Questionable Evaluation Ground Truth: The core evaluation of "embedding quality" relies on using DL-predicted binding affinities as ground truth. Given the well-known inaccuracies of current DL affinity predictors and the fact that real-world virtual screening typically yields hit rates of 10-20%, reporting precisions as high as 68% is misleading. It primarily shows that the embeddings reconstruct the source model's predictions, not their ability to identify truly active compounds.

2. Unsubstantiated and Suspicious Performance Claims:

    The 37.3x speedup over fingerprints yet only 1.5x speedup over re-running encoders seems implausible. Rule-based fingerprints are designed for extreme speed, and neural networks of encoders cannot be efficiently deployed on CPUs. The author should justified this highly unusual claim.

    Also, Table 3 shows identical results (e.g., 20%, 4%, 2%) for different fingerprint methods and similarity metrics. This statistical uniformity is highly improbable and suggests a potential error in the experimental setup, data processing, or reporting, severely undermining the credibility of the baseline comparison.

3. Limited Technical Novelty: The work is an integration of existing tools: an existing encoder (DeepPurpose/MPNN), existing databases (ZINC22, GDB13), and an existing search library (FAISS). While the scaling effort is non-trivial, the paper does not introduce a novel algorithm, model, or foundational insight for the machine learning field. It functions more as a systems/engineering report.

4. Inadequate Comparison to State-of-the-Art: The paper does not sufficiently situate itself against other high-throughput screening systems (e.g., the recently published BIOPTIC system (https://pubs.acs.org/doi/10.1021/acs.jcim.5c00743)) that also operate on vast chemical spaces. The comparison is limited to basic fingerprints and does not discuss the trade-offs of its approach.

**Questions:**

1. Ground Truth Justification: Given the known unreliability of DL-predicted binding affinities, how do you justify their use as the primary ground truth for evaluating retrieval quality? Have you performed any validation on a smaller, experimentally-verified dataset to correlate your reported precision with real-world hit rates?

2. Clarification on Speedup Claims: Can you detail the exact procedure for the "37.3x speedup vs. fingerprints and 1.5x vs. re-running the encoder" benchmark? Why re-running neural network-based encoders is even faster than fingerprints?.

3. Explanation of Identical Baseline Results: The results for all fingerprint baselines in Table 3 are identical or nearly identical. This is statistically very unexpected. Can you explain this phenomenon and verify the correctness of your fingerprint generation and similarity calculation code?

4. Defining the ML Contribution: Beyond the scale of the resource, what is the specific machine learning contribution of this work? What new methodological insight does it provide for representation learning or large-scale retrieval that is not already enabled by the existing, integrated components (FAISS, DeepPurpose, etc.)?

---

### Official Review · Reviewer_6Kss · 2025-10-27

**Soundness:** 2
**Presentation:** 2
**Contribution:** 1
**Rating:** 2
**Confidence:** 4

**Summary:**

This paper introduces EmbedMol, a large scale molecular embedding dataset containing 11.2 billion embeddings from GDB13 and ZINC22. The embeddings are generated using the DeepPurpose model (2020), aiming to provide a resource for molecular similarity search and representation learning. The work lowers the computational barrier for large scale studies and includes an API and basic evaluations to showcase potential applications.

**Strengths:**

1. The authors’ effort in building such a dataset using their computational resources is commendable. And it might be have some kind of contirbution to the community who needs existing embedding representations for a molecule library.

**Weaknesses:**

1.  The contribution of this paper is very limited. I admire the author's effort on using existing deep learning models to get the embeddings for a large scale molecule dataset. However, such contribution is not enough for a top-tier machine learning conference like ICLR. This is a pure engineering work.
2. Only one DL method is used to generate embeddings, and it is a method in 2020, 5 years from now.
3. The runtime comparison in Section 6.3 is questionable. When comparing with fingerprint-based methods, the baseline does not precompute fingerprints, while EmbedMol uses precomputed embeddings—this naturally favors EmbedMol in terms of speed. Additionally, the comparison with property prediction models, where SMILES inputs are simply replaced with embeddings, seems artificial and lacks practical relevance.

**Questions:**

see weakness

---

### Official Review · Reviewer_pEei · 2025-10-30

**Soundness:** 2
**Presentation:** 2
**Contribution:** 1
**Rating:** 2
**Confidence:** 3

**Summary:**

This paper introduces EmbedMol, an open billion-scale dataset of precomputed molecular embeddings, intended as a practical and reproducible resource for large-scale virtual high-throughput screening and evaluation of molecular representations. Experimental results demonstrate speedups and improved retrieval precision over traditional fingerprint baselines, while supporting large-scale retrieval tasks previously infeasible at this scale.

**Strengths:**

1. The dataset construction represents a solid engineering effort, achieving an impressive scale of over 11 billion molecules, which is uncommon in open molecular datasets.

2. The authors provide a reproducible pipeline for dataset generation, including implementation details and AWS-based automation scripts.

3. The resource may be useful for practitioners who need ready-to-use molecular embeddings for large-scale virtual screening, without extensive compute requirements.

**Weaknesses:**

1. The paper’s main contribution is infrastructural rather than scientific. It focuses on assembling and releasing a large dataset of precomputed molecular embeddings and the associated AWS-based generation pipeline. While this is a valuable engineering effort, the paper explicitly states that no new embedding model or methodological innovation is introduced. Consequently, the work does not advance our understanding of molecular representation learning, embedding design, or algorithmic properties. This makes it less aligned with ICLR’s emphasis on novel modeling ideas and theoretical insight, rather than infrastructure or resource releases alone.

2. All embeddings in the dataset are derived from a single pretrained MPNN encoder. Although MPNNs are well-established for molecular graphs, relying on a single architecture restricts diversity in representational properties and risks propagating the biases of that specific model family. Section 3 indicates no ablation or comparative study across alternative encoders. As a result, it remains unclear whether the dataset can serve as a general-purpose benchmark or whether its utility is confined to the embedding characteristics of this one model.

3. The evaluation protocol uses drug–target interaction scores predicted by a separate DTI model as “ground truth” for measuring retrieval precision. While this is pragmatically necessary at scale, it undermines the scientific validity of the results: retrieval performance may simply reflect the inductive bias of the prediction model rather than true biochemical relevance. Since no experimental validation with empirical assay data is provided, the reported precision metrics should be interpreted with caution.

4. The evaluation mainly contrasts EmbedMol with rule-based fingerprints such as Morgan or MACCS, which are now considered classical baselines. There is no comparison with more modern, learned molecular embeddings. This limits the ability to assess whether EmbedMol actually advances the state of molecular representation learning beyond existing deep models.

**Questions:**

The dataset currently relies on embeddings generated from a single pretrained MPNN encoder. Could the authors elaborate on how they expect the reported efficiency and retrieval quality to generalize to other architectures?

---

### Official Review · Reviewer_djuA · 2025-11-01

**Soundness:** 2
**Presentation:** 2
**Contribution:** 2
**Rating:** 4
**Confidence:** 2

**Summary:**

In this paper, the authors present EmbedMol, which is a dataset provides precomputed molecular embeddings. Their dataset contains 128-dimensional embedding vectors extracted using a deep learning-based encoder, so that users don't have to preprocess and encode SMILES strings every time. Their encoder uses drug-protein binding assay data, aiming to capture not only chemical structures but also some bioactivity-related patterns. The evaluation result shows that the search speed was up to 37x faster than traditional fingerprint-based search methods and approximately 1.5x faster than running the encoder every time.

**Strengths:**

EmbedMol is designed to facilitate fast and efficient drug discovery and molecular screening by providing a pre-computed 128-dimensional embedding vector. The evaluation results show that search and inference speed improve. The paper is easy to follow and well organized. The dataset is publicly available, which gives high reproducibility.

**Weaknesses:**

1. The paper doesn’t clearly show the breakdown of each step. It would be helpful if the authors reported how much time embedding and preprocessing each take.
2. It seems that using EmbedMol may prevent users from applying different deep learning models (non MPNN). If fine-tuning is possible with the dataset, could you please show how it converges compared to training from scratch, and demonstrate its generalizability by evaluating it on other tasks or models?
3. How does it compare to existing approximate nearest neighbor searching algorithms?

**Questions:**

Please see weakness section.

---

### Note · Authors · 2025-12-03

**Comment:**

To the Area Chair and Reviewers,

We thank the reviewers for their thoughtful comments and constructive feedback. After careful consideration, we have decided to withdraw our submission from ICLR.

Best Regards,
The Authors

**Withdrawal Confirmation:**

I have read and agree with the venue's withdrawal policy on behalf of myself and my co-authors.